# Captivity and the co-diversification of great ape microbiomes

Alex H. Nishida [1,2 ✉] & Howard Ochman[1,2]

Wild great apes harbor clades of gut bacteria that are restricted to each host species. Previous research shows the evolutionary relationships among several host-restricted clades mirror those of great-ape species. However, processes such as geographic separation, host-shift speciation, and host-filtering based on diet or gut physiology can generate host-restricted bacterial clades and mimic patterns of co-diversification across host species. To gain insight into the distribution of host-restricted taxa, we examine captive great apes living under conditions where sharing of bacterial strains is readily possible. Here, we show that increased sampling of wild and captive apes identifies additional host-restricted lineages whose relationships are not concordant with the host phylogeny. Moreover, the gut microbiomes of captive apes converge through the displacement of strains that are restricted to their wild conspecifics by human-restricted strains. We demonstrate that host-restricted and co-diversifying bacterial strains in wild apes lack persistence and fidelity in captive environments.

[1] Department of Integrative Biology, The University of Texas at Austin, Austin, TX, USA. [2] Molecular Biosciences, The University of Texas at Austin, Austin, TX, USA. ✉email: ahnishida@utexas.edu

Humans and other great apes harbor gut microbiomes whose compositions are distinct among hosts[1–4], with multiple bacterial lineages that are found only in a single species[5]. The presence of host-restricted taxa does not necessarily imply taxa are specific or specialized to hosts and can be traced to several processes, which occur within the complex microbial communities contained by hosts. For example, host species can differentially filter certain microbes from their environment based on their diet and gut physiology, thereby generating host-restricted microbial lineages even when bacteria are acquired horizontally anew each generation[6]. Host-restricted lineages can also result from co-diversification (or co-speciation), which refers to the situation where the host and symbiont lineages diverge in parallel and, as a result, exhibit congruent phylogenies[7,8]. In contrast to host filtering, co-diversification relies on vertical (e.g., parent to offspring) transmission between generations[9]. Often confused and used interchangeably with co-diversification, phylosymbiosis describes a pattern of congruence between overall microbiome composition similarity and host phylogeny, but is agnostic to which of these forces explain the distributions of microbial taxa within a complex community[7]. Co-diversification sometimes denotes co-evolution, in which there are reciprocal changes in the bacteria and host in response to their interactions, but it can also arise from vicariance without reciprocal selection between host and symbiont[9]. Note that host restriction need not denote a long-term association between bacteria and hosts, and can be generated when a symbiont colonizes and diversifies in a new host species well after the two hosts have diverged (sometimes referred to as host-shift speciation[8]). Host-switch events can sometimes mimic patterns of co-divergence when related bacterial strains colonize sister species, but they can also generate host-restricted microbial clades that erode patterns of co-divergence. Due to their similar outcomes, differentiating among these processes that underlie the appearance of host-restricted strains is often difficult, but it is necessary for understanding the forces shaping gut microbiome assembly.

The mammalian gut microbiome is highly diverse with relatively few lineages that show evidence of being vertically transmitted from mother to infant[10,11]; however, there are still some taxa that appear to have co-diversified with great apes. Because the 16S rRNA gene is unable to distinguish among closely related bacterial strains and species, it is not surprising that, when microbiomes are interrogated with 16S-amplicon rRNA sequencing, very few bacterial taxa show evidence of being co-diversified with primate host species[12]. To that end, by sequencing the more quickly evolving and more variable gyrase B gene, Moeller et al.[5] identified taxa not identified by 16S analysis[12] belonging to Bacteroidaceae and Bifidobacteriaceae families that show evidence of co-diversification with great ape species.

A way in which the host-specificity of co-diversified lineages can be evaluated is by analyzing the gut microbiomes of captive great apes. Captive individuals are separated from their ancestral populations, fed alternative diets, and exposed to bacteria harbored by caretakers and by other species living in close proximity, any of which could act to disrupt the host-microbe associations of their wild conspecifics. Under such restrictive conditions, bacterial strains or species that typically inhabit a host can be lost and unrecoverable[13], being displaced by novel bacteria[14] and/or by related strains occupying a similar functional niche[15]. In primates, captivity is consistently associated with lower gut microbial diversity and large taxonomic shifts, typically making them more similar to humans[16–18]. Yet, despite these sometimes major compositional changes, captive primates species, like their wild relatives, remain distinguishable based on their gut microbiomes[17,19]. Maintenance of host-restricted bacterial taxa despite dietary and environmental changes, and where inter-species transmission is possible, is compelling evidence of host specialization and co-evolution.

In this study, we survey wild apes, captive apes, and humans by profiling both 16S rRNA and the more highly variable gyrase B protein-coding gene to determine the dynamics of host-restricted and co-diversified bacterial lineages among captive apes. When examining captive apes, co-diversified bacterial lineages can have three potential fates: (1) they can persist in the captive conspecifics, suggesting that they co-evolve with hosts, (2) they can transfer to different host species, suggesting that geographic factors generated the original signal of co-diversification, or (3) they can be displaced by other microbial taxa, suggesting that host-filtering occurs based on changes in diet. We find that the increased sampling strategy adopted in this study identifies additional host-restricted lineages that are more consistent with host-shift diversification events rather than strict co-diversification, and exposes the determinants of microbiome diversification among great ape species.

## Results

**Increased sampling disrupts co-diversified lineages.** The gyrB amplicon data from wild apes ($n = 130$), captive apes ($n = 72$), and industrialized humans ($n = 16$) generated in this study and from published data[5], along with gyrB sequences extracted from metagenomic data from over 9000 humans worldwide[20] (Supplemental data 2), yield a total of 7596 ASVs that typed to the Bacteroidales order. Of these, 6784 are restricted to a particular host species, and assembling these gyrB-ASVs into the largest possible monophyletic clades produces 356 well-supported clades whose constituents are present in five or more host individuals (Fig. 1).

Given the differences in sampling among hosts, we considered only those host-restricted clades present in >25% of individuals of the various sample types (i.e., captive chimpanzees, wild gorillas, industrialized humans, etc.). Our increased sampling of wild apes and humans reveals that the co-diversifying lineages reported in Moeller et al.[5] represent only a subset of Bacteroidales diversity present in wild apes and humans. There are 26 host-restricted clades from wild apes that were previously not recognized and that do not coincide with the co-diversified lineages identified by Moeller et al.[5] (Fig. 1).

The addition of scores of host-restricted clades to the Bacteroidales phylogeny allowed re-examination of the lineages originally reported to co-diversify with great-ape species. Two of the original co-diversifying lineages now contain a diversity of Bacteroidales sequences from humans that disrupt the previous pattern of co-diversification (Fig. 2, Fig. S1). And in other cases, the addition of ASVs belonging to a different host species creates mixed-host clades that disrupt the congruence between the host and bacterial phylogenies. For example, in Bacteroidaceae lineage 2, the emergence of a host-restricted bonobo clade from within a co-diversifying chimpanzee clade results in the formation of two separate chimpanzee clades (Clade 2 in Fig. 2). Similarly, wild gorillas and wild chimpanzees harbor some closely related ASVs, and this mixed-host clade (Clade 4 in Fig. S1) splits the two co-diversifying clades of Bacteroidaceae lineage 3. Thus, the newly identified host-restricted clades that appear with increased sampling are better explained by bacterial diversification following multiple host-switch events.

Overall, we find that with additional sampling, most lineages previously described as co-diversifying no longer present topologies consistent with strict co-speciation and now contain a mixture of both host-restricted and mixed-host clades. Although three separate statistical tests provided significant support for co-diversification between the hominid phylogeny and Bacteroidaceae lineage 2 (PACo, $p < 0.001$; Parafit, $p = 0.001$; HCT, $p < 0.001$), these tests also reached the same level of support

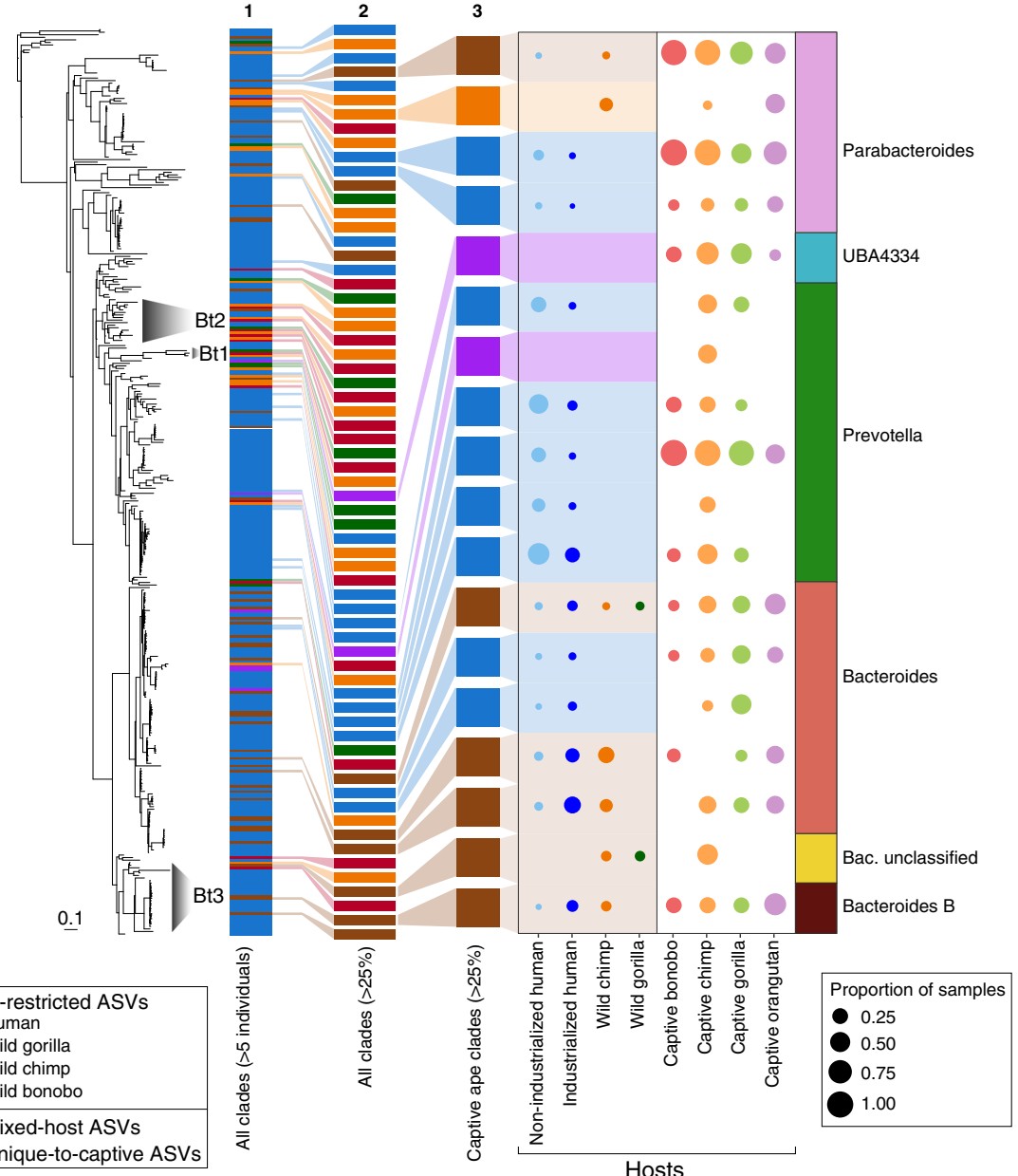

**Fig. 1 Host-restricted clades of wild apes are lacking in captive apes.** Phylogeny of host-restricted clades based on *gyr*B-ASVs in the order Bacteroidales. In the phylogeny to the left, the three labeled and shaded clades (Bt1, Bt2, Bt3) correspond to the co-diversified lineages of Bacteroidaceae identified in Moeller et al.[5]. Colors in the numbered columns that follow indicate the host-species source of ASVs constituting each clade, with column 1 displaying the sources of all 356 clades present in at least five individuals, column 2 displaying only those 65 clades present in >25% of either wild or captive individuals of any host species, and column 3 displaying only those 18 clades present in >25% of captive individuals of any host species. Circles in the next two columns are shaded according to host-species source and sized to indicate the proportion of samples harboring the clade. For each of the clades prominent in captive apes (column with white background), bacterial family and genus taxonomic assignments are color-coded in the final column.

when the host phylogeny was randomized (PACo, *p* < 0.001; Parafit, *p* = 0.001; HCT, *p* < 0.001). Cumulatively, these tests establish that *gyr*B-ASVs assort into host-restricted clades more often than expected by chance—but the fact that random host trees also produce significant host-restricted associations indicates that distance-based statistical tests are unreliable for determining whether the topology of host-restricted clades is consistent with co-diversification.

**Loss and sharing of wild ape host-restricted ASVs in captivity.** We initially set out to determine whether the co-diversified

bacterial lineages present in wild apes persist in captive apes, which would both demonstrate the fidelity of transmission despite major changes in lifestyle and geography, and lend support to the view that these bacterial lineages are important to, and likely co-evolved with, their host species. However, most host-restricted *gyr*B clades in wild-ape species, both those previously identified by Moeller et al.[5] and by this study, are absent from captive apes (Fig. 1). In the single case where we observe a wild chimp-host-restricted clade of wild chimpanzees that is also present in captive apes, it is no longer confined to chimpanzees but found in both captive chimpanzees and orangutans. Unlike wild apes, captive great apes largely harbor human-restricted,

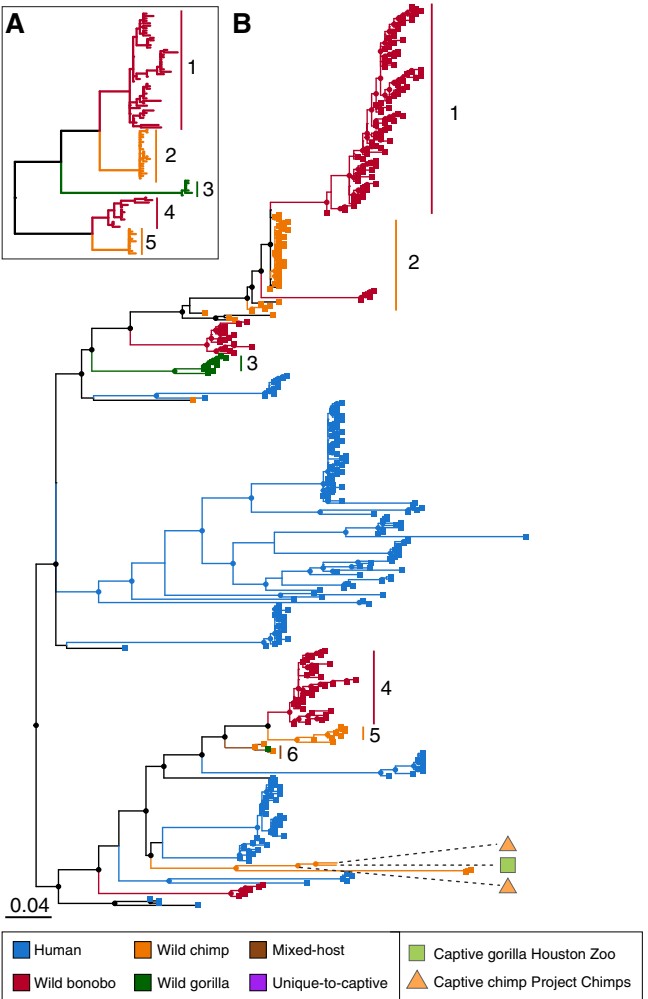

**Fig. 2 New ASVs within the co-diversified lineage Bacteroidaceae 2 (Bt2 in Fig. 1).** Inset (**A**) at the upper left shows topology of co-diversified clades originally identified by Moeller et al.[5], with branches color-coded to denote host species. The five major clades from this inset tree (Clades 1–5) are highlighted and labeled in the phylogeny (**B**), which includes the newly identified *gyrB*-ASVs. Lineages and terminal tips are color-coded to indicate the host-species source of an ASV, and dashed lines correspond to ASVs identified in captive apes. An emergent mixed-host clade is also labeled (Clade 6). Nodes with bootstrap support >50 are indicated.

Legend: Human | Wild chimp | Mixed-host | Captive gorilla Houston Zoo | Wild bonobo | Wild gorilla | Unique-to-captive | Captive chimp Project Chimps

mixed-host, and unique-to-captive-apes *gyrB* clades that are shared broadly among host-species.

To determine whether the transmission of *gyrB*-ASVs between captive apes and humans are recapitulated across broader taxonomic groupings, we examined the distribution of 16S-ASVs among wild apes ($n = 330$), captive apes ($n = 87$), industrialized ($n = 140$), and non-industrialized humans ($n = 134$) using microbiome composition data generated by this study as well as by other published studies[2,3,17,21–23] (Supplemental data 1). Relative to the *gyrB* dataset, a far greater proportion of Bacteroidales 16S-ASVs are identified as mixed-host (Fig. S2), which likely reflects the inability of the V4 region of the 16S rRNA gene to distinguish among closely related bacterial strains. However, despite the lower proportion of host-restricted ASVs in the 16S dataset, we are still able to examine their distribution across captive apes to test whether patterns observed in the *gyrB* dataset are consistent when a broader taxonomic diversity of microbial taxa is analyzed.

Paralleling the host distributions observed with fine-grained *gyrB* data, we find that host-restricted 16S-ASVs that are confined to a particular ape species in the wild are largely absent from the microbiomes of captive apes (Fig. 3B) (Kruskal–Wallis: wild bonobo *vs.* captive apes, df = 1, H = 127.1, p < 0.001; wild chimp *vs.* captive apes, df = 1, H = 144.8, p < 0.001; wild gorilla *vs.* captive apes, df = 1, H = 151.3, p < 0.001). Of the few wild-ape host-restricted 16S-ASVs that persist in captivity, the majority are present in multiple captive species. There are only three 16S-ASVs, two in gorillas and one in chimpanzees, that are exclusively present in wild and captive conspecifics and no other host species.

Instead of harboring strains that are present in their wild conspecifics, a large fraction of the captive-ape microbiome is composed of 16S-ASVs that are otherwise restricted to humans, consistent with a pattern of colonization by human-associated strains. In fact, the proportion of human-restricted 16S-ASVs observed in captive apes does not differ significantly from that in humans (Fig. 3B) (Kruskal–Wallis, df = 1, H = 6.4, p = 0.093). Even among those bacterial genera that are most common in wild apes and humans (Fig. 3C), the compositional shifts in captive apes are caused by an increase in human-restricted and mixed-host 16S-ASVs (Fig. 3D).

It is possible that the identification of ASVs as being host-restricted results from limited sampling, and, therefore, the findings that captive apes lack of wild-ape ASVs and possess human ASVs are affected by the relative sampling of wild apes, captive apes, and humans. However, this is not the case: Firstly, the identification of ASVs as host-restricted is not likely to be an artifact of sampling given that they display a similar relationship between mean relative abundance and prevalence across host samples as mixed-host ASVs (Fig. S3). Host-restricted ASVs are more common in the Bacteroidetes phylum than in the relatively less abundant Firmicutes phylum (df = 16, $X^2 = 255.93$, p < 0.0001) (Fig. S4), indicating there is a taxonomic pattern to host-restriction rather than a random distribution. Secondly, if captive apes were harboring unidentified strains that are present in their wild ape conspecifics, we expect to observe many unique-to-captive-ape ASVs that are restricted to a particular captive host species. However, ASVs observed only in captive apes are not more likely to be limited to a particular host species or site (Fig. S5). Lastly, if the frequency of human-restricted ASVs in captive apes were an artifact of under-sampling wild apes, we might expect that additional sampling of wild-ape populations would shift some human-specific ASVs to mixed-host ASVs. However, we found that wild apes usually possess mixed-host ASVs that are present only in other ape species, and that captive apes tend to harbor mixed-host ASVs that are present both in humans and in wild apes (Fig. 2), indicating that additional sampling of wild apes is not likely to uncover troves of additional 16S-ASVs that are disproportionately shared with humans.

**No evidence of host-species filtering in captive great apes**. A distinctive feature of the present study is the ability to compare the gut microbiomes of the same captive great-ape species from multiple locations, allowing us to determine the extent to which microbiome compositions adjust to each host species in captivity. If host species differentially filter bacterial strains, captive apes of the same host species are expected to harbor more similar microbiome compositions after controlling for zoo site and enclosure. Among captive apes, zoo site explains 31% of the variation in microbiome composition assessed by Bray-Curtis distance (PERMANOVA, df = 4, F = 10.6, p < 0.001, $r^2 = 0.32$; betadisper, df = 4, F = 5.3, p = 0.004), enclosure explains an additional 18% of the variation in microbiome composition

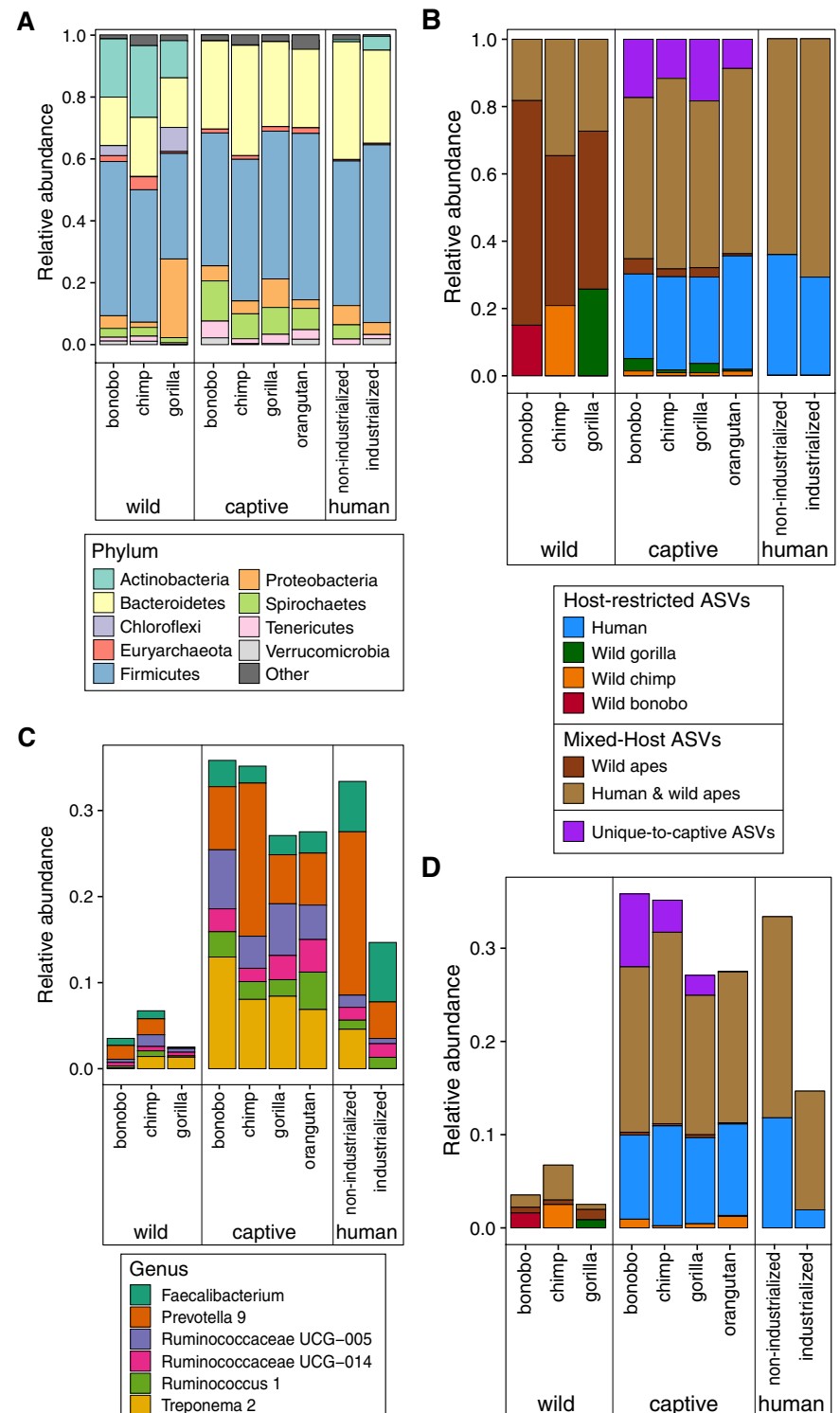

**Fig. 3 Compositional changes in captive great apes are associated with the loss of host-restricted ASVs from wild apes and the gain of host-restricted ASVs from humans.** Average relative abundances of (**A**) bacterial phyla (**B**) host-restricted 16S-ASVs, mixed-host 16S-ASVs, and unique-to-captive apes 16S-ASVs in the gut microbiomes of captive apes, wild apes, and humans from industrialized and non-industrialized societies, (**C**) bacterial genera, and (**D**) host-restricted 16S-ASVs, mixed-hosted 16S-ASVs, and unique-to-captive apes 16S-ASVs within the genera shown in (**C**). Note that average relative abundances of several genera listed in (**C**) and of the host-restricted 16S-ASVs, mixed-hosted 16S-ASVs, and unique-to-captive apes 16S-ASVs within these genera show parallel increases in all captive ape species affecting convergence in microbiome compositions.

(PERMANOVA, df = 5, $F$ = 4.9, $p < 0.001$, $r^2$ = 0.18; betadisper, df = 9, $F$ = 2.3, $p$ = 0.024), and host species is not significantly associated with microbiome composition after controlling for site and enclosure. Because we include data from studies that employed diverse methods to extract and sequence samples, we tested the degree to which study source contributed to the similarity of microbiome composition (after controlling for site and enclosure) but find no significant effect.

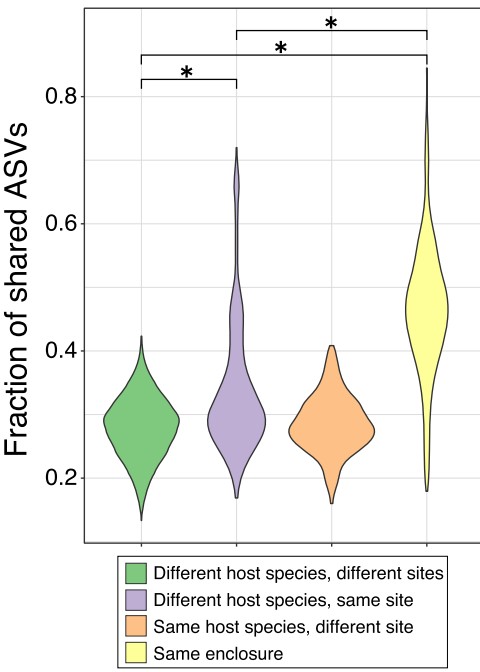

**Fig. 4 Cohabiting captive apes exhibit the highest levels of ASV sharing.**
Proportions of shared 16S-ASVs (Sørenson similarity index) determined by
in captive apes in relation to geography and host species membership.

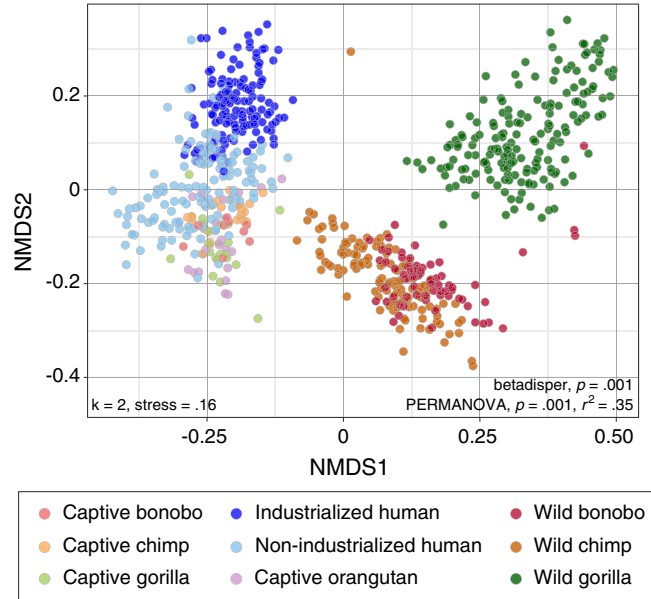

**Fig. 5 Gut microbiome convergence in captive great apes based on 16S-amplicon sequencing.** Gut microbiome compositions of captive great apes, wild great apes, and humans partitioned by species and lifestyle, and visualized by non-metric multidimensional scaling (NMDS) based on Bray-Curtis distances (PERMANOVA, df = 8, F = 47.9, p = 0.001, $r^2$ = 0.35; betadisper, df = 8, F = 44.0, p = 0.001).

We also compare the relative effects of shared geography, host species, and enclosure on the fraction of shared 16S-ASVs (i.e., Sorenson distance, which disregards the relative abundance of ASVs). Individual apes that are neither of the same host species nor residing at the same zoo site share a prodigious 30% of their 16S-ASVs (Fig. 4). This high degree of sharing among captive apes is indiscriminate of species assignment: there is no significant association between host species and 16S-ASV sharing after excluding individuals residing in the same enclosure (permutation *t*-test, df = 1, t = 2.2, p = 0.12), similar to what is observed when applying Bray-Curtis distance metrics. Shared geography is associated with a slight increase in the proportion of shared 16S-ASVs among individuals (permutation *t*-test, df = 1, t = 16.3, p = 0.012); however, this increase is due largely to the extensive sharing of 16S-ASVs among gorillas and chimpanzees at the Houston Zoo (Fig. S6). As expected, captive apes in the same enclosure (which are invariably the same species) exhibit the highest proportions of shared ASVs (±50%) (Fig. 4), far exceeding the influence of shared geography or host-species (permutation *t*-test, all comparisons, df = 1, t > 15.9, p = 0.012).

**16S microbiome composition of great ape species converge in captivity.** Based on our sampling of great apes from multiple locations, captivity disrupts gut microbiomes in a similar manner across host species (PERMANOVA, df = 8, F = 47.9, p < 0.001, $r^2$ = 0.35; betadisper, df = 8, F = 44.0, p < 0.001). The gut microbiomes of captive apes are more similar to those of other captive host species than to their wild ape counterparts (Fig. 5; Supplemental data 5), a convergence due to parallel increases in the relative abundance of the Bacteroidetes (Kruskal–Wallis: captive vs. wild chimp, df = 1, H = 46.5, p < 0.001; captive vs. wild bonobo, df = 1, H = 16.5, p = 0.002, captive vs. wild gorilla, df = 1, H = 29.0, p < 0.001) and Spirochaetes (Kruskal–Wallis: captive vs. wild chimp, df = 1, H = 28.1, p < 0.001; captive vs. wild bonobo, df = 1, H = 16.5, p = 0.002, captive vs. wild gorilla, df = 1, H = 35.9, p < 0.001) and decreases in the relative

abundances of Actinobacteria (Kruskal–Wallis: captive vs. wild chimp, df = 1, H = 62.7, p < 0.001; captive vs. wild bonobo, df = 1, H = 23.6, p < 0.001, captive vs. wild gorilla, df = 1, H = 56.0, p < 0.001) (Fig. 3A).

The reduction in the abundance of Actinobacteria in captive apes is accompanied by a reduction in actinobacterial diversity; however, total bacterial diversity in captive apes, though varying by zoo site, is similar to that of wild apes (Fig. S7). All captive ape species have increased abundances of six bacterial genera, including multiple genera of Ruminococcaceae that are common in the human microbiome (Fig. 3C; Supplemental data 6). Based on all metrics tested, captive ape microbiomes are more similar to those of humans living in non-industrialized societies (Fig. 5 and Supplemental data 5; Bray-Curtis distance; Fig. S8, Jaccard, unweighted and weighted UniFrac distances). The similarity is due, in part, to the prevalence of *Treponema*, which are rarely present in humans sampled from industrialized societies (Fig. 2B). Captive apes also exhibit a loss in the relative abundances of microbial genera that are unique to and distinguish the microbiomes of wild ape species (Fig. S9). For instance, *Acinetobacter* is characteristic of the microbiome of wild gorillas but is absent from captive gorillas. These compositional shifts are consistent across captive-ape host species sampled from multiple sites and studies (Fig. S10). Overall, captivity homogenizes great ape gut microbiomes such that individuals exhibit a loss of microbial taxa characteristic of wild apes and a gain of taxa prominent in humans.

**Discussion**

The microbiomes of great apes in captivity have provided new insights into the fidelity and persistence of bacterial lineages that are restricted to ape species in the wild. Based on analyses of both 16S-ASVs and the more highly variable *gyrB*-ASVs in wild apes, captive apes, and humans, we show that the gut microbiomes of great ape species in captivity change and converge due to both the loss of strains present in their wild conspecifics and the gain of

strains that are restricted to humans. Apes in captivity do not exhibit evidence of filtering based on host species after controlling for site and enclosure. Through the increased sampling of captive apes and the inclusion of data extracted from a broad range of metagenomic surveys, we identified several additional host-restricted strains and clades that disrupt the phylogenies of bacterial lineages previously reported to have co-diversified with their great ape hosts.

Captivity represents a natural laboratory that has helped establish whether bacterial strains restricted to a particular host species in the wild are capable of colonizing other host species when they inhabit the same environment. Of the few host-restricted strains detected in wild apes that remain in captive apes, both the wild-gorilla restricted and the wild-chimpanzee restricted 16-ASVs are present in multiple captive great ape species, not just in their corresponding conspecifics. Houtz et al.[24] argue that captive apes maintain host-specific taxa based on their finding that ASVs belonging to some microbial genera are more similar to wild counterparts than to humans; however, their analysis did not examine whether these ASVs were present in other captive great ape species. Therefore, patterns of host restriction detected in wild ape hosts do not reflect a biological barrier to host colonization, and at least on contemporary timescales, there is little evidence that host species differentially filter strains. Even captive great apes of the same species do not share a higher proportion of ASVs, except when individuals reside in the same enclosure, indicative of a large social component to the transmission and sharing of bacterial strains. Higher ASV sharing among apes in the same enclosure may also reflect differences in the environmental bacteria across small distances, as microhabitat differences have also been observed in wild primate populations[25].

Microbiome compositions of great apes in captivity converge regardless of host species or geographic location and become similar to humans, as has been shown previously[17,26,27]. This contrasts the situation in wild apes in which microbiome compositions are specific to host species even when sampled from locales where multiple species are sympatric and sharing the same seasonal regimes[4,28]. One potential source of the difference between wild and captive apes is that species in captivity consume similar diets of primate chow supplemented with the cultivated fruits and vegetables common to human diets, whereas wild ape species differentially forage even when residing in the same geographic region[29]. Although captive apes often live in proximity to industrialized humans (i.e., in urban zoos), the microbiomes of captive great apes are always more similar to those of non-industrialized humans[26,30]. Great ape species in captivity harbor increased frequencies of *Treponema* and *Prevotella*, which are correlated with the metabolism of complex carbohydrates in non-industrialized human populations[31–33], and contrasts the situation in humans, in which industrialization is associated with the loss of *Treponema* and *Prevotella*[31,34,35]. This shift in captive apes is not anticipated given that they generally consume far less dietary fiber than do wild populations[36].

Aside from compositional changes in the microbiome that are associated with captivity, as has been the focus of previous studies[17,18], we examined whether these shifts are driven by the expansion or loss of endogenous wild-ape strains or by colonization by human strains. When captive individuals possess bacterial genera that are common in humans but absent from wild populations[17], it is most likely that the human-associated strains were acquired while in captivity. However, assessing bacteria at this taxonomic rank provides little information about their source and transmission. By examining variation at finer taxonomic levels, i.e., individual ASVs, we found that captive apes typically harbor human-restricted strains, which have excluded or replaced wild-ape restricted strains of the same genus (e.g., *Prevotella* and *Treponema*). That taxa exhibiting the greatest increases in relative abundance in captive apes are derived from humans may indicate

that the human-restricted strains are better adapted to the domesticated gut conditions and diet of captive apes. An alternate explanation is that these strains are encountered more regularly by both humans and captive apes based on shared geography. Other such cases of strain displacement have previously been reported[37,38]; for example, certain strains of *Prevotella* have a greater capacity for complex carbohydrate metabolism in humans with a vegan diet[32].

The loss of wild-ape host-restricted strains by captive apes in multiple locations, including those strains belonging to lineages previously identified as co-diversifying with their great ape hosts, indicates that host-strain restriction and transmission do not readily persist in captivity. Because the majority of apes included in this study were born in captivity and their source populations not sampled, we cannot trace the specific wild-ape host-restricted strains that have been lost. However, we can assume that those bacterial lineages that typically colonize and diversify with all great ape species would have been present in the ancestors of captive apes, and their loss, exclusion, or displacement could result from the dietary and lifestyle changes associated with captivity. Aside from losses in bacterial diversity known to be associated with reduced dietary fiber[13,23,39], social contacts influence the maintenance and transmission of bacterial lineages among troop members in the wild[40,41], and the limitations imposed by captivity[42] could have downstream effects on microbiome diversity.

We identified several new host-restricted lineages of gut bacteria, underscoring the effects of sampling when determining patterns of host restriction and co-diversification between hosts and their microbes. Re-examination of the co-diversified lineages of Bacteroidaceae with increased sampling yielded several additional lineages that disrupted the co-diversification pattern originally reported in Moeller et al.[5]. These new bacterial lineages were by-and-large restricted to individual host species, but their phylogenetic relationships were not fully consistent with the branching order of their hosts. Therefore, the co-diversification patterns reported by Moeller et al.[5] probably reflect bacterial lineages that arose and diversified in geographically isolated host species, thereby producing host-restricted clades. But there being only 3–4 host species of great apes, the topology of select subtrees of host-restricted clades may, by chance, resemble that of the host phylogeny. We applied a randomized host phylogeny to show that in cases in which there are many closely related ASVs restricted to different host species, it is easy to generate a false positive co-diversification result. Therefore, we advise caution about using these statistical tests used to infer co-diversification between mammals and their microbiome constituents: While these tests can handle multiple associations between host and symbiont, there are not designed or tested with the very large number of associations present in host-microbiome relationships.

Aside from co-diversification and co-evolution, there are several circumstances that can lead to the advent of host-restricted bacterial clades: (i) dispersal and host-shifts, in which a bacterial lineage diversifies following colonization of a different host species, (ii) anagenesis or duplication, in which the bacteria speciate without speciation of the host, (iii) extinction and/or displacement of the bacterial lineage[8,9]. It is likely that Bacteroidales, though obligately anaerobic[43], are subject to enough dispersal in the environment to colonize alternate host species and are maintained in populations long enough to diversify into distinct clades before being lost or replaced.

By tracing the distribution of the bacterial strains maintained by wild ape populations in their captive conspecifics, we demonstrate that host-restricted clades are readily lost in captivity. The lack of persistence and fidelity of these host-restricted strains in their host species suggests that, at least among lineages

within the Bacteroidales, there are no strong fitness dependencies or strong host-selectivity between symbionts and their hosts. By pairing observations of wild and captive populations, we report a great contrast between the ready transmission of bacterial strains among captive ape species and many newly reported host-restricted clades among wild ape species. Our results suggest that patterns of host restriction are likely to be the result of bacterial dispersal and diversification among isolated host populations rather than co-speciation.

## Methods

**Sample collection**. Great-ape fecal samples, including bonobos ($n = 13$), chimpanzees ($n = 26$), gorillas ($n = 22$), and orangutans ($n = 11$) (Supplemental data 1) were obtained from the Houston Zoo, Columbus Zoo, and Project Chimps (Blue Ridge, GA), and subjected to 16S (Supplemental data 1) and gyrase B (*gyrB*) amplicon analysis (Supplemental data 2). We complied with ethical regulations for animal testing and research, with approval from AZA boards granted at each location. Fecal samples (50–100 g) were collected within 12 h of deposition and stored at $-20\,°C$ in RNA*later* (Invitrogen). Sampling took place over the course of a single day to minimize the chance individuals were represented twice. Due to the manner of collection, each sample derives from one individual within a known social group residing in a shared enclosure; however, the source identity of a given sample was in many cases unknown. Individual histories and dietary information for captive great apes were provided by the Houston and Columbus Zoos (Supplemental data 3). All individual hosts from the Houston Zoo, with the exception of two chimpanzees, were born in captivity; and in the samples from the Columbus Zoo, all individuals, with the exception of four bonobos, were born in captivity.

**16S microbial community profiling**. DNA extractions from captive ape fecal samples were performed both by protocols specified by the Earth Microbiome Project[44] and by a phenol-chloroform-isoamyl alcohol (25:24:1), bead-beating procedure[45]. Initial characterization of microbial communities assessed variation in the V4 region of the 16S rRNA gene (Supplemental data 4). Sample DNA was amplified in triplicate using the conventional 515F and 805R primers designed for dual-index barcoding and sequencing according to the manufacturer's instructions (Illumina). PCRs were cleaned with AMPure XP beads (Beckman Coulter), and barcoded amplicons were pooled and sequenced on the Illumina MiSeq and iSeq platforms.

To expand representation of human gut microbiomes, we combed publicly available databases to select human samples representing both pre-industrial[22,23] ($n = 134$) and industrialized societies[21,23] ($n = 140$) to correspond roughly with the number of wild apes gorillas ($n = 176$), chimpanzees ($n = 85$), and bonobos ($n = 69$) that were included from Moeller et al.[2] (Supplemental data 1). In addition, we augmented representation of non-human great apes by including data from studies that evaluated the gut microbiomes of captive great apes ($n = 15$)[3,17] (Supplemental data 1). To avoid a potential source of bias among datasets[46], only those studies that assayed the V4 region of the 16S rRNA gene were included.

**16S data processing**. Raw sequence data, whether produced for the present study or procured from public databases, were processed in the identical manner as follows: Multi-sample fastqs were demultiplexed (Supplemental information for program details), and adaptor and primer sequences were removed using Cutadapt[47]. For each dataset, the resulting sequences were quality-filtered, error-corrected, chimera-cleared filtered, and datasets with pair-end reads were merged using DADA2 to generate Amplicon Sequence Variants (ASVs)[48]. Merged paired-end sequences were then trimmed to a length of 250 bp so that sequences could be combined with datasets that provided single-end 250 bp reads. ASVs were taxonomically assigned using the SILVA reference database (version 132)[49], and the resulting ASV table, taxonomy table, and metadata were imported into Phyloseq[50]. ASVs assigned to chloroplast or mitochondrial sequences, or not assigned to either Bacteria or Archaea, were removed. To minimize the influence of low-abundance taxa on measures of alpha diversity, ASVs not reaching 0.5% relative abundance in at least two samples were also removed. Samples were rarefied to a depth of 10,000 reads to generate the final ASV table used in subsequent analyses. Abundance-filtered sequences were aligned with mafft[51], and phylogenies generated with Fasttree[52].

**Microbiome composition based on 16S ASVs**. Microbial diversity estimates were calculated with several metrics, including Faith's Phylogenetic diversity, number of ASVs, and Inverse Simpson, using the picante[53] and Phyloseq[50] R packages. We assessed whether sample groups exhibited significant differences in microbial diversity using Kruskal–Wallis and Dunn multiple comparisons tests with FDR corrected *p*-values in the PMCMR R package[54]. To evaluate how alpha-diversity patterns differ among bacterial phyla, we performed comparisons among the number of observed ASVs belonging to Firmicutes, Bacteroidetes, Proteobacteria, and Actinobacteria individually. To quantify differences in microbiome composition among samples, beta diversity was measured by Bray-Curtis and Jaccard

dissimilarity indices, and by weighted and unweighted UniFrac distances. Compositional differences among samples were visualized through NMDS ordination. The relative influences of captivity status and host species were tested by PERMANOVA with 999 permutations and differences in the homogeneity of groups were tested using betadisper in the Vegan R package[42].

To further investigate trends observed in the NMDS ordination, we determined distances between group centroids by conducting pairwise PERMANOVAs among all possible combinations of the nine groups—wild bonobos, wild chimps, wild gorillas, captive bonobos, captive chimps, captive gorillas, captive orangutans, non-industrialized humans, and industrialized humans. To examine shifts in microbiome composition between wild and captive apes, differences in the relative abundances of microbial genera were assessed using two-sided Kruskal–Wallis and Dunn multiple comparisons tests with Bonferroni-corrected *p*-values that accounted for the number pairwise comparisons for each genus.

**Host-species filtering by 16S ASVs**. Whereas previous studies of captive apes typically cluster 16S amplicons into 97% OTUs, we analyzed individual 16S-ASVs to gain finer resolution of variant distribution. We determined whether ASVs in captive apes were unique to specific enclosures, zoos, or host species, or were shared across multiple host species and/or locations. We employed a multiple PERMANOVA to assess the relative influence of site, enclosure, host species, and dataset on Bray-Curtis distance between samples. We calculated the extent to which ASVs were shared between pairs of individuals according to their fraction of shared ASVs by Sørenson similarity. We tested which samples had higher fractions of shared ASVs (due either to recovery from the same host species or the same zoo) using a two-sided permutation *t*-test with Bonferroni-corrected *p*-values.

**Defining host-restricted 16S-ASVs**. Host-restricted ASVs are those that are exclusively observed in a single host species; but due to the potential for strain transmission between apes in captivity, our designation of host-restricted clades or ASVs considers only those confined to humans or to any one wild-ape species. ASVs found in multiple wild ape host species and/or humans were designated as "mixed-host". ASVs not found in wild apes or humans but observed only in captive conspecifics were designated as "unique-to-captive". We assessed the proportions of host-restricted, mixed-host, and unique-to-captive ASVs across host species and captivity status by averaging the relative abundances of ASVs across all individuals within a sample group.

**gyrB-amplicon sequencing and library preparation**. In addition to assessing microbial diversity via 16S amplicon analysis, we assayed sequence variation in a 250-bp region of the single-copy protein-coding gene gyrase B (*gyrB*)[5,55]. In this study, we profiled the Bacteroidaceae diversity of captive and wild apes by sequencing a portion of the *gyrB* gene, sampled from captive bonobos ($n = 13$), captive chimpanzees ($n = 26$), captive gorillas ($n = 22$), captive orangutans ($n = 11$), wild chimpanzees ($n = 21$) and wild gorillas ($n = 18$) (Supplemental data 2). This region evolves at a much faster rate than 16S rDNA[55], allowing us to differentiate ASVs labeled as mixed-host by 16S analysis into host-restricted ASVs. Because the rapid rate of *gyrB* evolution precludes fabrication of universally conserved primers, we targeted the *gyrB* region of Bacteroidales, the most prevalent bacterial order in the gut microbiome of captive apes (Supplemental data 4). Primer design, library preparation, and PCR reaction conditions, and cleanup were performed as previously described[55], and the resulting barcoded amplicons were pooled and sequenced.

**Processing gyrB data**. Similar to 16S data processing, adaptor and primer sequences were removed from the *gyrB* raw sequence data with Cutadapt and quality-filtered using DADA2[48]. In addition, we augmented the *gyrB* dataset with previously published data from wild great apes sampling gorillas ($n = 19$), chimpanzees ($n = 48$), bonobos ($n = 24$), and industrialized humans ($n = 16$)[5].

**Specifying reference taxa**. To assign taxonomy to *gyrB* ASVs, we created a custom *gyrB* reference dataset using reference genomes from the GTDB-Tk database[56]. We note that because GTDB-Tk normalizes the taxonomic rank of genomes, taxonomic assignments can differ from those of other databases (e.g., NCBI; SILVA). For instance, GTDB-Tk considers *Prevotella* and other related genera to be within the family Bacteroidaceae whereas the NCBI and SILVA databases recognize them as within the Prevotellaceae family. Genomes from the Bacteroidetes phylum along with representative genomes from other phyla were annotated using prodigal[57], and *gyrB* sequences were extracted using the GTDB-Tk *gyrB* hmm profile (TIGR01059.HMM) with hmmer3[58]. Only full-length hits with an E-value < 1e−250 were retained. Because *gyrB* sequences observed in wild apes diverge from the reference *gyrB* sequences, we assigned taxonomy to *gyrB* ASVs using translated sequences with IDTAXA[59] through the DECIPHER R Package[60].

**Processing metagenomic samples**. To explore the extent of *gyrB* variation in human gut microbiomes, we extracted *gyrB* sequences from the extensive set of metagenomic assemblies produced by Pasolli et al.[20], downloaded from the Segata Lab data repository (http://segatalab.cibio.unitn.it/data/Pasolli_et_al.html)

(Supplemental data 2). We combined *gyrB* ASVs with all Bacteroidales *gyrB* reference sequences to generate a query fasta that was used to conduct a blastn search of >9000 human metagenomic assemblies. We used the blastn results to extract *gyrB* sequences that exactly overlapped amplified regions from metagenomic assembly fastas.

**Merging amplicon and metagenomic *gyrB* datasets**. We formatted metagenomic samples and their associated *gyrB* sequences into a sequence table that could be merged with *gyrB* amplicon data using DADA2. Because metagenomic and amplicon sequencing interrogate communities differently, we did not consider the relative abundance of *gyrB* ASVs, and instead considered only the presence or absence of ASVs in samples. Amplicon and metagenomic samples included in the *gyrB* analysis are listed in Supplemental data 2. As an additional filtering step, ASVs were translated in frame, and those sequences containing premature stop codons, or that aligned with <80% amino-acid identity or with <90% alignment length to the *gyrB* reference database described above were removed. *gyrB* ASVs were aligned based on their corresponding amino acid sequences with the DECIPHER R package[60]. The phylogeny of *gyrB* ASVs along with the *gyrB* reference sequences was constructed with Fasttree[52].

**Defining host-restricted clades with *gyrB***. We determined the proportions of individual *gyrB*-ASVs that are host-restricted using the same approach applied to 16S-ASVs. For *gyrB*-ASVs, we also analyzed the distribution of host-restricted ASVs by grouping closely related variants into host-restricted clades. We define a host-restricted clade as a monophyletic group of variants (i.e., ASVs or *gyrB* variants) that are exclusive to individuals of the same host species. We note again that our designation of host-restricted clades or ASVs only considers those confined to humans or any of the wild-ape species, allowing us to determine whether captive apes have acquired human-associated variants or share host-restricted variants in their captive environment.

We iterated over the *gyrB* amplicon phylogeny using the Python package ete3[61] to find the largest possible monophyletic clades of host-restricted ASVs having bootstrap support >50 and present in ≥5 individuals. We selected these thresholds because random permutation of the ASV matrix showed that host-restricted clades of this size were unlikely to be produced in wild-ape host species by chance. The ASV matrix was permuted 999 times using a binary null model, preserving row and columns frequencies, with the permatfull function in the Vegan R package[42].

After defining host-restricted clades, we determined whether remaining ASVs could be clustered into well-supported mixed-host clades using the methods described above for host-restricted clades. Finally, we determined whether ASVs that do not fall into either host-restricted or mixed-host clades are observed in well-supported clades that were unique to captive apes. After defining clades as either host-restricted, mixed-host, or unique-to-captive apes, we compared the frequency and taxonomic distribution of each clade in wild apes and determined which were prominent in captive ape species. We used blastn to identify all ASVs that match with >95% identity to the co-diversified clades reported in Moeller et al.[5] and extracted subtrees that represented the most recent common ancestors of ASVs matching these co-diversifying clades within each of the three lineages. On these subtrees, we visualized (i) host-restricted clades, (ii) host-restricted ASVs (which may or may not fall into host-restricted clades), (iii) ASVs present in captive apes ranked according to host species and location, and (iv) ASVs within the co-diversified clades identified by Moeller et al.[5].

**Co-speciation tests**. We conducted multiple co-speciation tests, including Parafit[62], PACo[63], and Hommola et al.[64] using 999 permutations on the Bacteroidaceae 2 lineage with either the great ape host phylogeny or a random host phylogeny. Host species divergence times were based on median times obtained from TimeTree[65]. All three approaches are distance-based tests that utilize both a host and a microbial dissimilarity matrix along with a matrix detailing all host-microbe associations. Code for implementing and comparing the performance of Parafit[62], PACo[63], and Hommola et al.[64] was from modified from Balbuena et al.[63] (https://www.uv.es/cophylpaco/index.html).

**Reporting summary**. Further information on research design is available in the Nature Research Reporting Summary linked to this article.

## Data availability

16S-amplicon and *gyrB*-amplicon sequence data produced by this study are publicly available under accession numbers ("PRJNA692991"; "PRJNA693013"), and we used accession numbers listed in Supplemental data 1 from published studies to acquire additional data.

## Code availability

Code and processed data to generate all figures and tables are available in a dedicated Github repository (https://github.com/ahnishida/captive_ape_microbiome, https://doi.org/10.5281/zenodo.5188501).

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

## Acknowledgements

We would like to thank the following people and organizations for providing fecal samples from captive primates: Judy McAuliffe and Houston Zoo staff, Audra Meinelt, Laura Pierson and Columbus Zoo staff, and Project Chimps staff. We also thank Steven Kyle for his help in sequence library preparation, and Marian Schmidt, Paul Kirchberger, and Tobin Hammer for discussions and comments on earlier drafts of this manuscript. Funding for this research was provided by the National Science Foundation (Graduate Research Fellowship 2016226761 to A.H.N.) and the National Institutes of Health (Award Number R35GM118038 to H.O.).

## Author contributions

A.H.N. and H.O. conceived and designed the study; A.H.N. analyzed data; A.H.N. and H.O. wrote the manuscript.

## Competing interests

The authors declare no competing interests.
