## [Peer Review File · Nature Communications]

REVIEWER COMMENTS

Reviewer #1 (Remarks to the Author):

This paper tests the extent to which microbial strains are ape host-species specific by comparing unique and shared strains in captive African apes (and orangutans) and wild African apes, as well as humans. It demonstrates that many of the species-specific strains identified in wild apes are lost in captive apes, and that captive apes acquire microbial strains typically associated with humans. It also re-tests the extent to which microbial strains co-diversify with hosts, as originally reported by Moeller et al. 2016, and finds that co-diversification is not robust to increased sample size and geographic coverage.

Overall, the findings are interesting. I think the most novel finding is that strains from the wild are replaced by those from humans. While long assumed, no study has yet demonstrated this empirically. Some of the other findings are less novel (convergence of captive species' microbiomes, lack of host-microbe co-diversification), but they provide additional valuable data to the existing literature.

My main criticism of this paper is that the introduction does not provide sufficient foundation for the subsequent material. First, the theoretical underpinning is weak. There are key gaps in the explanation of potential processes underlying patterns of phyllosymbiosis, and current knowledge is not sufficiently described. There are a number of papers on phyllosymbiosis more generally and primate microbiomes more specifically that are not cited or discussed. Additionally, hypotheses and predictions are not clearly laid out, making it difficult to follow some of the methods and analyses. For example, lineages shared across very specific groups of samples are utilized in the analyses (host-restricted, mixed-species, etc.), but why these grouping were chosen and what they allow us to test (with regard to the theory the paper begins with) are not explicitly described in the introduction. While readers can figure this out after getting through the whole paper, the paper would be much stronger if the set up were stronger and made more of these links explicitly.

Another important detail that would strengthen the manuscript is a clearer indication of what results are based on the 16S data (all ASVs) versus the gyrB data (Bacteroidaceae) throughout. In general, the authors move back and forth between hypotheses, results, and interpretations that rely on both of these data types, and sometimes it blurs the distinction between findings that were true overall versus specific to Bacteroidaceae. For example, the abstract and introduction make no mention of a specific subset of microbial lineages (Bacteroidaceae) despite the fact that half of the data from the analyses are particular to this subset of lineages. More clarity about which conclusions

are generalizable across all ASVs versus just the Bacteroidaceae would improve readability and impact.

Other specific comments follow:

Line 21: Amato et al. 2019, ISME J also demonstrates this for apes within the entire Order. This should be cited.

Line 27: There is a third option not mentioned here despite the fact that it is believed to be one of the most important processes leading to patterns of host-specific microbiomes. Selection for specific microbes by host genetics/physiology can result in this pattern. This does not necessarily have to be linked to dispersal limitation or host-microbe co-diversification. These ideas are reviewed in Kohl 2020, Phil Trans Soc B. Further the Amato et al. 2019 paper above reports that >90% of primate-associated microbiomes are not co-diversifying with hosts and that, in agreement with Kohl and others, there appear to be host physiological filters that are selecting for specific microbiomes. All of this should be explored/introduced.

Line 37: I think one more dot needs to be connected in the explanation of logic here. The idea is that if strains are lost, they are not co-diversifying/important to the host, correct? This is implied but not explicitly stated. The next paragraph does this to some degree, but in general these two paragraphs need to more clearly lay out the hypothesis/predictions to set up the experiment.

Line 64: Were these samples collected on a single day to ensure individuals were not represented twice in the dataset? Please state.

Line 66: Please put sample size information in the main text.

Line 77: Please briefly name the datasets and cite in the main text.

Line 79: What wild apes? None have been described for this paper yet.

Line 81: Again, please identify and cite datasets in the main text. Citations are necessary at the very least.

Line 92: Were only forward reads used?

Line 101: Were the effects of study tested or controlled for in any of these statistical analyses? I would recommend this. This is also part of the reason it would be helpful to indicate the number of wild/captive populations and where they came from. These study-associated factors could bias the results depending on how the different sample types were distributed across studies, field sites/zoos, etc.

Line 118: I am not sure if it makes sense to have captive orangutans in this analysis if there are no wild orangutans unless there are some analyses that only consider patterns within captive samples. Again, clarifying the premise and study design earlier will help with this.

Line 131: It seems that strains unique to a host species and present in both wild and captive individuals would represent another important group of microbial lineages to identify. These are likely to be the microbes that are essential to some aspect of host physiology/fitness (or are at least being selected for by host physiology) since they are retained across environments. I see this implied later in the methods but seems to be missing here. Also, as previously mentioned, laying out these different groups in the introduction and explaining their theoretical utility would be helpful.

Line 162: There are other wild ape metagenomes available in the Amato et. al. paper that could contribute to this analysis further as well and reduce geographic bias in the wild data.

Line 206: Again, seems to me like ASVs found in a given host species regardless of environment should be included here. They would be the most likely to be co-diversifying in my mind.

Line 211: Data do not appear to be public yet.

Line 212: Please list the accession numbers and cite here.

Line 215: What does 'in a uniform manner' mean? Were pairs of wild/captive groups compared across species?

Line 343: I think it is worth emphasizing that this pattern is among closely-related host species (i.e. African apes). It is also interesting to me that in the UniFrac ordination plots in Fig S2, microbiome similarity still appears to track host phylogeny in captivity...this suggest there is still some sort of host selection occurring. Discussing this subtlety would be helpful. It may relate to the specificity of some results to Bacteroidales, which is not mentioned until later.

Line 357-362: These patterns have also been reported by Clayton et al. 2016 (already cited previously) and Amato et al. 2015, Microbiome. Reduced fiber diets in primates appear to result in increased Prevotella, unlike what is observed in humans. These studies should be cited here.

Line 391: Also, exposure to environmental microbiomes, particularly in early life. I suggest the authors consider a recent paper by Narat (2021, Scientific Reports) that emphasizes that the environmental exposures of apes in captivity are not always the same, and differences by zoo, etc. may affect the microbial patterns observed. I know different captive environments are included in this analysis, reducing the potential impact of bias, but I think it is still an important point to raise. Additionally, an interesting question here is whether the wild-born captive apes showed any difference patterns from the other captive individuals. I realize the sample size is small, but this could still provide some insight into the importance of early life environments versus current social interactions and/or diet.

Line 394: This is the first time Bacteroidales are mentioned in either the intro or the discussion, despite the fact that many of the findings are limited to this taxon. I suggest clarifying this point in both the introduction and discussion (currently better addressed in the discussion).

Line 400: Again, mention of the fact that most microbial lineages do not appear to co-diversify with primate hosts would be appropriate here (per Amato et al. 2019).

Reviewer #2 (Remarks to the Author):

In this paper, Nishida and Ochman examine the gut microbiomes of wild and captive apes, comparing them with various human populations. They ask if captive apes retain host-specific strains of bacteria identified in wild populations and/or if there is displacement of those strains by human-specific strains. Additionally, they examine if the patterns of co-divergence of Bacteroidales found in earlier work hold up with more in-depth sampling. The questions asked are interesting, the methodology is sound, and the authors' results support their findings that there is convergence in captive ape gut microbiomes and that bacterial lineages that were previously described as co-diversifying may not be. These results are consistent with recent primate microbiome work, and are based on 16S and gryB amplicon sequencing, neither of which are new methods - most recent studies examining co-diversification and/or strain-level profiling of microbial communities use shotgun metagenomic sequencing. I think the introduction and discussion could use some strengthening, as well.

Major comments:

The introduction could be more comprehensive and at times is unclear or confusing. In the introduction, co-adaptation, co-evolution, and co-diversification are all mentioned, and the distinctions between these terms is unclear. For example, I find the sentence in lines 24-27 a bit of an overstatement and misleading. Co-evolution (as defined in the cited Groussin paper and elsewhere) is simply reciprocal evolutionary changes between two organisms in response to the selective pressures of each. It does not necessarily result in the "most advantageous combination," simply one of many possible combinations that increase fitness relative to the last combination.

What do you mean by "dependence of particular bacteria-host combinations" in line 32?

The humanization of captive/domesticated microbiomes is not a novel finding (see Clayton et al., 2016; Reese et al., 2021 in eLife; Tsukayama et al., 2018 in mSystems). Additionally, recent articles have suggested that the patterns of dissimilarity in human and ape microbiomes do not necessarily follow a pattern of phylosymbiosis (Gomez et al., 2019 in mSphere; Amato et al., 2019 in 2019 in Genome Biology) and there is limited evidence of co-diversification in primate-associated microbial lineages (Amato et al., 2019 in ISME). The information is missing from the introduction and discussion.

I appreciate the amount of detail used when describing the bioinformatic analysis. It still would be very helpful to include a link to a publicly available repository where the code used in the analyses can be found. This would be especially helpful for the gryB analyses.

Do you still see species-specific microbial communities when examining only the captive ape species? This information would help to contextualize the captive vs. wild ape PERMANOVA results.

Not only would captive apes of the same host species be expected to harbor larger proportions of shared ASVs than captive apes of different host species, but they would perhaps be expected to differentially acquire human-restricted ASVs. Are there differences in which human-restricted ASVs are found in each captive ape species or not?

Lines 344-347: The higher rates of shared ASVs within enclosures than between enclosures indicates not only a large social component to bacterial transmission, but also environmental transmission. It is unlikely that any two enclosures at a zoo have the environmental microbes (due to differences in enrichment devices, access to outdoor areas, plants in outdoor areas, etc). We see these microhabitat differences even within geographic locations in the wild (for example, Grieneisen et al., 2019 in Proc B).

Lines 355-357: Reese et al., 2021 in eLife could additionally be cited here.

Lines 371-374: There is no definitive evidence in this paper that the human-restricted strains are better adapted to captive apes. They simply may be taxa that are encountered regularly by captive apes and for which there are no barriers to colonization in apes. This may be related to diet but could also be related to features of the bacteria themselves (virulence, ability to evade detection by the immune system, etc).

The paper lacks in its discussion of what we know about turnover of the gut microbiome and vertical transmission. The discussion section focuses on the role of diet, lifestyle, and sociality in loss of host-specific strains in captive apes, but neglects to substantially discuss whether the bacterial strains that have been lost are vertically or horizontally transmitted (ie, would they need to be reacquired from the environment or social contact or not?).

Minor comments:

Line 81: Should this refer to wild great apes instead of captive great apes? Looking at Table S1, it appears all samples from other studies are either wild apes or humans.

What length sequencing reads were used and were they paired or single end?

Line 156: Bacteroidetes is misspelled here.

Figure 2a: Is this relative to all sequences or just bacterial sequences? If it is the later, why do the relative abundances not sum to one? Should there be an "other" category?

Reviewer #3 (Remarks to the Author):

In this study, Nishida and Ochman compare the presence of bacterial lineages in great apes in the wild with those in captive environments, and with industrialized and non-industrialized humans, using both 16S rRNA and *gyrB* based ASVs. The authors report host restricted bacterial lineages of wild apes are shared among multiple host species in captivity. Additionally, the microbiomes of all great ape species converge in captivity through the acquisition of lineages restricted to non-industrialized humans. There are some important findings here, the most important being that the authors do provide new evidence that questions much of the co-specification of bacterial lineages with great apes and humans reported previously by the Ochman group. The authors provide further strong data that transmission between ape species is one of the dominant factors that shape the ape microbiome and thus disrupts patterns of co-diversification. The importance of under-sampling of the previous study is also discussed. Unfortunately, the paper is rather chaotic, the major findings do not become clear in the abstract (which focuses on the detection of human lineages in captive apes), and the authors decided to put way too much focus on findings that are clearly confounded due to variation in the methodology, sampling, and under-sampling.

Major points:

Writing: The abstract is vague and key findings, such as the disproval of co-speciation, is barely mentioned. Instead, focus is put on new findings that might also be just the result of under-sampling. Focus on the writing should be put on the absence of co-speciation as this is an important

finding. Authors should further consider submitting a Correction to Science as their previous finding on co-speciation made substantial impact in the field but seem to be false. Throughout the manuscript, the authors refer to strain-level analysis, although none of their methods achieve this. For me, this manuscript would have to be completely re-written and focused on the major findings (no co-speciation, ecology of captive species and the importance of transmission between ape species for gut microbiota ecology) while acknowledging that other findings on shared lineages with humans are speculative, confounded. Limitation of the study should also be discussed (see below).

Limitations: The study has serious limitations. There are only very few captive ape populations being studied and they are compared with very few populations of wild apes. It is therefore possible that lineages present in captive apes are just absent in wild apes as they are under-sampled. It is a bit inconsistent that the authors conclude that one of their major previous findings is the result of under-sampling, while then never discussing the limitations of their new findings. Additional confounders are the different DNA-extraction methods (kit versus phenol-chloroform based), different data-sets from different studies (all of which used different methods, different people doing the work, different sequence runs, etc.), use of sub-optimal analysis methods that, despite the claims being made, are not able to differentiate strains.

Specific points

Title and abstract are vague. Be clear that you provide data that supports the importance of transmission between host species both in captivity, but also in the wild, that question the co-diversification of bacterial lineages in hominids.

Lines 9-10: 'Strains' are not assessed in this work, nor would they co-evolve.

Lines 16-18: This is not clear.

Line 123: An ASV-based analysis using 16S rRNA sequences is not at strain-level. It is often not even species level. This is consistently wrong throughout the manuscript.

Lines 214ff and Figure 1. This analysis is confounded as different DNA extraction methods and different data sets are used. None of the confounders have been considered in this analysis. Please provide data on PERMANOVA using these parameters. Zoo and population should also be included.

Line 245: The methods used here do not determine strains.

Line 255: Same as line above.

Line 272: Same as line above. I guess you get the point. Do not refer to strain throughout manuscript.

Lines 315-319: This is an important finding, but as presented, it is vague and should be expanded. So, after considering all the analysis, how many of the lineages identified as co-diversified with host species are still valid?

Lines 324-326: It is questionable that the authors conclude how under-sampling dis-validated their previous findings, but then never consider that their new study now might suffer from exactly the same problem. This might have led to some of the rather confusing findings (e.g. shared types between captive apes and only non-industrialized humans).

Dear referees,

We thank the reviewers for their comprehensive feedback on our manuscript. To fully address their comments and concerns, we have substantially revised and re-organized the manuscript, and conducted numerous additional analyses. Each of the points raised by the reviewers is enumerated listed below (reviewer's comments in italics, responses in Roman), but the major changes are summarized as follows:

- I. We restructured the manuscript and reanalyzed certain data in order to highlight (i) the fate of co-diversifying Bacteroidaceae lineages, (ii) the replacement in captive apes of ape-restricted strains by human-restricted strains, and (iii) the lack of host-specific filtering in captive apes. This shifts the focus away from our less novel findings (e.g., about compositional changes in gut microbiomes, which have been covered by others). Additionally, the introduction and abstract have been revised to improve clarity and to provide additional appropriate background.
- II. We performed new analyses to address questions raised by reviewers. This included additional PERMANOVA analyses (i) to investigate the extent to which host species contributes to variance in microbiome composition and (ii) to ask if the variation in sample size and numbers contributes to variance in microbiome composition. Additionally, we implemented several new co-speciation tests, including PACo, HCT, Parafit.

REVIEWER COMMENTS

Reviewer #1:

1. My main criticism of this paper is that the introduction does not provide sufficient foundation for the subsequent material.

We have overhauled and substantially revised the introduction to address these concerns.

2. the theoretical underpinning is weak. There are key gaps in the explanation of potential processes underlying patterns of phyllosymbiosis, and current knowledge is not sufficiently described.

The Introduction has been revised to expand on mechanisms underlying patterns of host-restricted bacterial taxa, and now includes discussion of host-filtering, co-diversification, and host-shift speciation (Lines 20-36). We do not include phyllosymbiosis because the patterns that it encompasses are not statistically distinguishable from those being evaluated, and the focus of the paper is on the distribution of host-restricted bacterial taxa and co-diversification.

3. Additionally, hypotheses and predictions are not clearly laid out, making it difficult to follow some of the methods and analyses. For example, lineages shared across very specific groups of samples are utilized in the analyses (host-restricted, mixed-species, etc.), but why these grouping were chosen and what they allow us to test (with regard to the theory the paper begins with) are not explicitly described in the introduction.

The revised manuscript clearly lays out our hypotheses (Lines 62-66) and has been edited to clarify the distinctions between hypotheses, and the rationale underlying the analyses, results, and interpretations.

4. Another important detail that would strengthen the manuscript is a clearer indication of what results are based on the 16S data (all ASVs) versus the gyrB data (Bacteroidaceae) throughout. In general, the authors move back and forth between hypotheses, results, and interpretations that

rely on both of these data types, and sometimes it blurs the distinction between findings that were true overall versus specific to Bacteroidaceae.

In restructuring of the results, we revised the text and figure legends to clarify which conclusions are generalizable across both datasets, and which are specific to either 16S or *gyrB*.

5. *Line 21: Amato et al. 2019, ISME J also demonstrates this for apes within the entire Order. This should be cited.*

We have added the relevant background conveying the lack of co-speciating lineages found in primates as reported in Amato et al. 2019 and also contrast those findings with patterns of co-diversification reported by Moeller et al. 2016 (Lines 41-48).

6. *Line 27: There is a third option not mentioned here despite the fact that it is believed to be one of the most important processes leading to patterns of host-specific microbiomes. Selection for specific microbes by host genetics/physiology can result in this pattern. This does not necessarily have to be linked to dispersal limitation or host-microbe co-diversification. These ideas are reviewed in Kohl 2020, Phil Trans Soc B.*

This issue is addressed in our response to point 2.

7. *Further the Amato et al. 2019 paper above reports that >90% of primate-associated microbiomes are not co-diversifying with hosts and that, in agreement with Kohl and others, there appear to be host physiological filters that are selecting for specific microbiomes. All of this should be explored/introduced.*

This issue is addressed in our responses to points 2 and 5.

8. *Line 37: I think one more dot needs to be connected in the explanation of logic here. The idea is that if strains are lost, they are not co-diversifying/important to the host, correct? This is implied but not explicitly stated. The next paragraph does this to some degree, but in general these two paragraphs need to more clearly lay out the hypothesis/predictions to set up the experiment.*

This is addressed in our response to point 3.

9. *Line 64: Were these samples collected on a single day to ensure individuals were not represented twice in the dataset? Please state.*

We address this comment, stating that, "Sampling took place over the course of a single day to minimize the chance individuals were represented twice" (Lines 76-78).

10. *Line 66: Please put sample size information in the main text.*

We have added sample-size information for both the *gyrB* and 16S datasets to the Methods section (Lines 93-97, 163-164, 174-175) as well as the main text (Lines 247-248, 298-299).

11. *Line 77: Please briefly name the datasets and cite in the main text.*

This issue is addressed in our response to point above.

12. *Line 79: What wild apes? None have been described for this paper yet.*

We have clarified that the wild apes included in this study are from previously published data for the 16S dataset (Lines 92-96).

13. *Line 81: Again, please identify and cite datasets in the main text. Citations are necessary at the very least.*

This issue is addressed in our response to point 10.

14. Line 92: *Were only forward reads used?*

We added information stating that some 16S datasets were paired-end and some were single-end reads (Lines 108-112).

15. Line 101: *Were the effects of study tested or controlled for in any of these statistical analyses? I would recommend this. This is also part of the reason it would be helpful to indicate the number of wild/captive populations and where they came from. These study-associated factors could bias the results depending on how the different sample types were distributed across studies, field sites/zoos, etc.*

We performed additional PERMANOVA analyses to examine the influence of study-associated factors (Lines 346-354). We show microbiome compositions of captive apes sampled by 3 different studies converge despite differences in sample collection and library preparation.

16. Line 118: *I am not sure if it makes sense to have captive orangutans in this analysis if there are no wild orangutans unless there are some analyses that only consider patterns within captive samples. Again, clarifying the premise and study design earlier will help with this.*

These samples are included because they help examine host-specific filtering of 16S-ASVs among captive great ape species.

17. Line 131: *It seems that strains unique to a host species and present in both wild and captive individuals would represent another important group of microbial lineages to identify. These are likely to be the microbes that are essential to some aspect of host physiology/fitness (or are at least being selected for by host physiology) since they are retained across environments. I see this implied later in the methods but seems to be missing here.*

We have expanded the Results section to discuss the limited number of ASVs that are exclusively found in wild-ape and captive-ape conspecifics for both the *gyrB* and 16S datasets (Lines 291-293; Lines 310-313).

18. Line 162: *There are other wild ape metagenomes available in the Amato et. al. paper that could contribute to this analysis further as well and reduce geographic bias in the wild data.*

The inclusion of additional metagenomic datasets, beyond the thousands that we have already considered, is not necessary to this manuscript revision. We have already uncovered numerous new host-specific lineages and shown the disruption of co-diversified lineages, and newly identified strains cannot revert these results.

19. Line 206: *Again, it seems to me like ASVs found in a given host species regardless of environment should be included here. They would be the most likely to be co-diversifying in my mind.*

This issue is addressed in our response to point 17.

20. Line 211: *Data do not appear to be public yet.*

The data have now been made public.

21. Line 212: *Please list the accession numbers and cite here.*

We include accession numbers in Tables S1 and S2.

22. Line 215: *What does 'in a uniform manner' mean? Were pairs of wild/captive groups compared across species?*

This sentence has been clarified in the Results section (Lines 369-372).

23. Line 343: *I think it is worth emphasizing that this pattern is among closely-related host species (i.e. African apes). It is also interesting to me that in the UniFrac ordination plots in Fig S2, microbiome similarity still appears to track host phylogeny in captivity...this suggest there is still some sort of host selection occurring. Discussing this subtlety would be helpful. It may relate to the specificity of some results to Bacteroidales, which is not mentioned until later.*

We conducted additional PERMANOVA analyses to show host-species is not associated with sharing of more ASVs after controlling for site and enclosure (Lines 341-351).

24. Line 357-362: *These patterns have also been reported by Clayton et al. 2016 (already cited previously) and Amato et al. 2015, Microbiome. Reduced fiber diets in primates appear to result in increased Prevotella, unlike what is observed in humans. These studies should be cited here.*

We added several additional citations regarding humanization of captive microbiomes (Lines 55-57; 420-421), and contrast the increased abundance of Prevotella and Treponema associated with captivity with decreases associated with human industrialization (Lines 430-435).

25. Line 391: *Also, exposure to environmental microbiomes, particularly in early life. I suggest the authors consider a recent paper by Narat (2021, Scientific Reports) that emphasizes that the environmental exposures of apes in captivity are not always the same, and differences by zoo, etc. may affect the microbial patterns observed. I know different captive environments are included in this analysis, reducing the potential impact of bias, but I think it is still an important point to raise.*

The reviewer raises an interesting point, but since we do not know the early environment of these individuals, it would be difficult to implement.

26. *Additionally, an interesting question here is whether the wild-born captive apes showed any difference patterns from the other captive individuals. I realize the sample size is small, but this could still provide some insight into the importance of early life environments versus current social interactions and/or diet.*

We cannot address this directly because, in many cases, we do not know which individual in an enclosure is the source of a fecal sample.

27. Line 394: *This is the first time Bacteroidales are mentioned in either the intro or the discussion, despite the fact that many of the findings are limited to this taxon. I suggest clarifying this point in both the introduction and discussion (currently better addressed in the discussion).*

In the revised version of the of Introduction, we clarify the role of Bacteroidales and *gyrB* sequencing in identifying patterns of co-diversification among wild apes, and contrast this with findings based on 16S sequences (Lines 41-48). Additionally, throughout the Results section, we highlight the distinction between findings based on the *gyrB* datasets specific to Bacteroidales and the 16S dataset capturing all bacterial taxa.

28. Line 400: *Again, mention of the fact that most microbial lineages do not appear to co-diversify with primate hosts would be appropriate here (per Amato et al. 2019).*

This issue is addressed in our response to point 5.

Reviewer #2 (Remarks to the Author):

1. *The introduction could be more comprehensive and at times is unclear or confusing.*

We restructured of the introduction; see response to Reviewer 1, Points 2–5.

2. *In the introduction, co-adaptation, co-evolution, and co-diversification are all mentioned, and the distinctions between these terms is unclear.*

We clarified our terminology and explanation of co-evolution and co-diversification (Lines 25-31).

3. *What do you mean by “dependence of particular bacteria-host combinations” in line 32?*

This issue is addressed in the point above.

4. *The humanization of captive/domesticated microbiomes is not a novel finding (see Clayton et al., 2016; Reese et al., 2021 in eLife; Tsukayama et al., 2018 in mSystems).*

This issue is addressed in our response to Reviewer 1, point 24.

5. *Additionally, recent articles have suggested that the patterns of dissimilarity in human and ape microbiomes do not necessarily follow a pattern of phyllosymbiosis (Gomez et al., 2019 in mSphere; Amato et al., 2019 in 2019 in Genome Biology).*

This issue is addressed in our response to Reviewer 1, point 2.

6. *There is limited evidence of co-diversification in primate-associated microbial lineages (Amato et al., 2019 in ISME). The information is missing from the introduction and discussion.*

This issue is addressed in our response to Reviewer 1, point 5.

7. *I appreciate the amount of detail used when describing the bioinformatic analysis. It still would be very helpful to include a link to a publicly available repository where the code used in the analyses can be found. This would be especially helpful for the gryB analyses.*

We provide a link to the github account containing all of the source code in the data accessibility section (Lines 244-245).

8. *Do you still see species-specific microbial communities when examining only the captive ape species? This information would help to contextualize the captive vs. wild ape PERMANOVA results.*

This issue is addressed in our response to Reviewer 1, point 23.

9. *Not only would captive apes of the same host species be expected to harbor larger proportions of shared ASVs than captive apes of different host species, but they would perhaps be expected to differentially acquire human-restricted ASVs. Are there differences in which human-restricted ASVs are found in each captive ape species or not?*

We show that there is no significant effect of host-species filtering after controlling for shared site and enclosure, so there is no expectation that specific species share specific ASVs with humans.

10. *Lines 344-347: The higher rates of shared ASVs within enclosures than between enclosures indicates not only a large social component to bacterial transmission, but also environmental transmission. It is unlikely that any two enclosures at a zoo have the environmental microbes (due to differences in enrichment devices, access to outdoor areas, plants in outdoor areas, etc).*

The reviewer makes a very good point in saying that microenvironments can vary among enclosures to enclosure (which might influence the similarity of apes residing in the same enclosure. Therefore, we have added this consideration to the Discussion (Lines 418-420).

11. *Lines 355-357: Reese et al., 2021 in eLife could additionally be cited here.*

We have added this citation, as requested (Line 421-422, 428-430).

12. *Lines 371-374: There is no definitive evidence in this paper that the human-restricted strains are better adapted to captive apes. They simply may be taxa that are encountered regularly by captive apes and for which there are no barriers to colonization in apes. This may be related to diet but could also be related to features of the bacteria themselves.*

We agree with the reviewer's critique and have edited the discussion to acknowledge adaptation is one of several possibilities (Lines 445-451).

13. *The paper lacks in its discussion of what we know about turnover of the gut microbiome and vertical transmission. The discussion section focuses on the role of diet, lifestyle, and sociality in loss of host-specific strains in captive apes, but neglects to substantially discuss whether the bacterial strains that have been lost are vertically or horizontally transmitted.*

We have revised and restructured the Introduction to explicitly discuss the role of transmission mechanisms in generating host-restricted bacterial taxa (Lines 22-28, 39-40).

14. *Line 81: Should this refer to wild great apes instead of captive great apes? Looking at Table S1, it appears all samples from other studies are either wild apes or humans.*

Captive apes from published studies were included in Table S1, although they are largely outnumbered from human and wild-ape samples. We re-ordered this table so that captive apes from published studies are reported after captive apes from this study.

15. *What length sequencing reads were used and were they paired or single end?*

We added information stating that some 16S datasets were paired-end and some were single-end reads (Lines 108-112). Trimmed lengths of sequenced reads used for our analyses is stated.

16. *Line 156: Bacteroidetes is misspelled here.*

The spelling has been corrected (Line 181).

17. *Figure 2a: Is this relative to all sequences or just bacterial sequences? If it is the later, why do the relative abundances not sum to one? Should there be an "other" category?*

We edited the figure to include an "other" category.

Reviewer #3 (Remarks to the Author):

1. *The paper is rather chaotic, the major findings do not become clear in the abstract (which focuses on the detection of human lineages in captive apes). The authors decided to put way too much focus on findings that are clearly confounded due to variation in the methodology, sampling, and under-sampling. The abstract is vague and key findings, such as the disproval of co-speciation, is barely mentioned. Instead, focus is put on new findings that might also be just the result of under-sampling. Focus on the writing should be put on the absence of co-speciation as this is an important finding.*

The reviewer is correct and their message is clear. As noted above, the Abstract and Introduction have been revised and now highlight major hypotheses and findings. Additionally, we reorganized the Results section to focus on (i) reconsideration of strictly co-speciating lineages in the *gyrB* data, (ii) replacement of wild ape-restricted ASVs by human-restricted ASVs in captive apes in both the *gyrB* and 16S datasets, and (iii) the lack of host-specific filtering in captive apes. These changes relegated our less novel findings (mostly about 16S microbiome composition) to the end. Note also that we demonstrate that replacement of wild ape-restricted ASVs by human-restricted ASVs in captive apes is not due to sampling (Lines 322-340).

2. *Authors should further consider submitting a Correction to Science as their previous finding on co-speciation made substantial impact in the field but seem to be false.*

Previous work drew its conclusions based on the data that were then available. “Corrections” involve ethical issues or the integrity of the work, and subsequent contrary findings to do not fall into that category (otherwise a very large number of papers, *e.g.*, every phylogenetic revision, would require a correction). Therefore, the present manuscript is, in a sense, the correction, and its publication demonstrates that increased sampling revises our understanding of the situation.

3. *Throughout the manuscript, the authors refer to strain-level analysis, although none of their methods achieve this.*

We refrain from the use of the term strain-level analysis for the 16S dataset. However, we think it is reasonable to infer that a single 16S-ASV denotes a group of closely related bacterial strains.

4. *For me, this manuscript would have to be completely re-written and focused on the major findings...while acknowledging that other findings on shared lineages with humans are speculative, confounded.*

Yes, indeed. As described above, the manuscript has been substantially rewritten.

5. *Limitations: The study has serious limitations. There are only very few captive ape populations being studied and they are compared with very few populations of wild apes. It is therefore possible that lineages present in captive apes are just absent in wild apes as they are under-sampled. It is a bit inconsistent that the authors conclude that one of their major previous findings is the result of under-sampling, while then never discussing the limitations of their new findings.*

In the revised, we devote a new section of the Results to demonstrate that replacement of wild ape-restricted ASVs by human-restricted ASVs in captive apes is not due to sampling (Lines 322-340).

6. *Additional confounders are the different DNA-extraction methods (kit versus phenol-chloroform based), different data-sets from different studies (all of which used different methods, different people doing the work, different sequence runs, etc.), use of sub-optimal analysis methods that, despite the claims being made, are not able to differentiate strains.*

This issue is addressed in our response to Reviewer 1, point 15.

7. *Title and abstract are vague. Be clear that you provide data that supports the importance of transmission between host species both in captivity, but also in the wild, that question the co-diversification of bacterial lineages in hominids.*

We have revised of Abstract to highlight our major findings. However, we disagree with the critique of the title, since it is descriptive of the diversity of findings presented by this study.

8. *Lines 9-10: ‘Strains’ are not assessed in this work, nor would they co-evolve.*

This issue is addressed in our response to Reviewer 3, point 6.

9. *Lines 16-18: This is not clear.*

This sentence has been revised (Lines 10-13).

10. *Line 123: An ASV-based analysis using 16S rRNA sequences is not at strain-level. It is often not even species level. This is consistently wrong throughout the manuscript.*

This issue is addressed in our response to Reviewer 3, point 6.

11. Lines 214ff and Figure 1. This analysis is confounded as different DNA extraction methods and different data sets are used. None of the confounders have been considered in this analysis. Please provide data on PERMANOVA using these parameters. Zoo and population should also be included.

This issue is addressed in our response to Reviewer 1, points 15 and 23.

12. Line 245: The methods used here do not determine strains.

This issue is addressed in our response to Reviewer 3, point 6.

13. Line 255: Same as line above.

This issue is addressed in our response to Reviewer 3, point 6.

14. Line 272: Same as line above. I guess you get the point. Do not refer to strain throughout manuscript.

This issue is addressed in our response to Reviewer 3, point 6.

15. Lines 315-319: This is an important finding, but as presented, it is vague and should be expanded. So, after considering all the analysis, how many of the lineages identified as co-diversified with host species are still valid?

We have restructured the Results section to highlight both our finding that additional sampling identifies many new host-restricted clades and that the new topology is more consistent with bacterial diversification following multiple host-switch events not coincident with host speciation (Lines 261-273). We also implemented several co-speciation tests (PACo, HCT, Parafit) on a previously identified co-diversified bacterial lineage and on the host phylogeny, as well as with a randomized host phylogeny (Lines 274-284).

16. Lines 324-326: It is questionable that the authors conclude how under-sampling dis-validated their previous findings, but then never consider that their new study now might suffer from exactly the same problem. This might have led to some of the rather confusing findings (e.g. shared types between captive apes and only non-industrialized humans).

This issue is addressed in our response to Reviewer 3, point 5.

REVIEWERS' COMMENTS

Reviewer #1 (Remarks to the Author):

The authors have substantially revised the manuscript and addressed my previous comments thoroughly. Although I continue to think that the findings represent a somewhat incremental advance in knowledge compared to the existing literature on host-specific microbiomes and microbiome shifts with captivity (and given the narrow microbial taxonomic breadth targeted), the framing and analyses are now much more robust. I note just a few minor points:

Line 42-46: this sentence is confusing

Line 65: I don't think this has to suggest co-evolution still

Line 67: For symmetry it would be good to indicate what this suggest

Line 153: "any one wild ape species" perhaps - this was confusing the first time I read it

309: typo? "Three, two"

Reviewer #2 (Remarks to the Author):

The substantial revision and refocus of the paper has improved it. Many of my concerns have been addressed, including all of my concerns about the discussion and methodological details. However, while the introduction is much improved, I still feel it is at times confusing and missing key information. Additionally, while the use of gryB in addition to 16S does provide some additional information on the codiversifying Bacteroidaeae lineages in captive populations, this paper's findings are similar to other studies of captive vs. wild primate gut microbiomes.

The authors set up the introduction by talking about the distinction between two processes (host filtering and codivergence) that lead to host-restriction (phylosymbiosis?) of microbial taxa and argue that it is important to determine when codiversification is occurring to better understand ecoevolutionary processes contributing to gut microbiome assembly (I agree!). They then go on to talk about whether or not codiversified lineages are host-specific – I think clarifying here that some host-restricted microbes are not necessarily host-specific would be helpful AND that they are hoping to test whether a taxa has actually codiversified or exhibits a pattern of codiversification by examining wild and captive individuals of a species. A key finding seems to be that these

Bacteroidaceae lineages seem to be codiversifying until you look at captive individuals. Additionally, it is not clear to me why coevolution of host-microbe pairs would result in codiversifying host-associated microbes persisting in captivity based on the information presented in the introduction. For hypotheses 1 and 2, the authors interpret what the results would mean for our understanding of processes underlying gut microbiome assembly, but do not do the same for hypothesis 3.

In the abstract and introduction, it still seems the concept of phylosymbiosis – “Previous research shows the evolutionary relationships among several of these host-restricted clades mirror those of great-ape species” – is being conflated with codiversification. These patterns are similar but distinct – phylosymbiosis only requires congruence in evolutionary relationships, whereas codiversification is a process by which phylosymbiosis can arise through a congruence in evolutionary history. See Davenport et al., 2017 in BMC Biology; Moran and Sloan, 2015 in PLoS Biology; and Mazel et al., 2018 in mSystems.

Additionally, the statistical tests for phylosymbiosis vs. codiversification are distinct. While methods examining only tree topology concordance detect phylosymbiosis, the methods employed in the current study (tests of cophylogeny) are specific to codiversification. Thus, there should be no harm in specifying that you are testing codivergence specifically and not phylosymbiosis.

A recently published study (Houtz et al., 2021 in Molecular Ecology) testing many of the same hypotheses as this study should be addressed in the intro and discussion:
<https://onlinelibrary.wiley.com/doi/abs/10.1111/mec.15994>

One smaller thing that might be helpful to add to the introduction - while vertical transmission in mammals might be rare, it seems that microbiome heritability may be high (see recent work here: <https://science.sciencemag.org/content/373/6551/181.full>).

Thank you for adding the link to the publicly available code!

REVIEWERS' COMMENTS

Reviewer #1 (Remarks to the Author):

1. *Line 42-46: this sentence is confusing*

We clarified the meaning of this sentence (lines 63-64).

2. *Line 65: I don't think this has to suggest co-evolution still*

Based on how co-evolution is defined and recognized, the presented arguments would lead one to conclude that there is co-evolution (lines 87-94).

3. *Line 67: For symmetry it would be good to indicate what this suggests*

We added a clause to the sentence (lines 93-94).

4. *Line 153: "any one wild ape species" perhaps - this was confusing the first time I read it*

We edited the sentence according to the reviewer's suggestion (line 449).

5. *Line 309: typo? "Three, two"*

We corrected this sentence (lines 162-166).

Reviewer #2 (Remarks to the Author):

6. *However, while the introduction is much improved, I still feel it is at times confusing and missing key information.*

We outline how we address the reviewer's concerns in points 7-9.

7. *I think clarifying here that some host-restricted microbes are not necessarily host-specific would be helpful AND that they are hoping to test whether a taxa has actually codiversified or exhibits a pattern of codiversification by examining wild and captive individuals of a species. Additionally, it is not clear to me why coevolution of host-microbe pairs would result in codiversifying host-associated microbes persisting in captivity based on the information presented in the introduction.*

We added a line specifying that host-restricted taxa are not necessarily specialized to host species (Line 37). We also a line clarifying how maintenance of host-restricted taxa in a captive environment is an indication of host specialization (84-86).

8. *For hypotheses 1 and 2, the authors interpret what the results would mean for our understanding of processes underlying gut microbiome assembly, but do not do the same for hypothesis 3.*

We added a clause to the sentence (Lines 93-94).

9. *In the abstract and introduction, it still seems the concept of phylosymbiosis – "Previous research shows the evolutionary relationships among several of these host-restricted clades mirror those of great-ape species" – is being conflated with codiversification. These patterns are similar but distinct – phylosymbiosis only requires congruence in evolutionary relationships, whereas codiversification is a process by which phylosymbiosis can arise through a congruence in evolutionary history.*

Additionally, the statistical tests for phylosymbiosis vs. codiversification are distinct. While methods examining only tree topology concordance detect phylosymbiosis, the methods employed in the current study (tests of cophylogeny) are specific to codiversification. Thus, there

should be no harm in specifying that you are testing codivergence specifically and not phylosymbiosis.

We added clarifying sentences that distinguish between co-diversification and phylosymbiosis, and have cited several additional studies (lines 45-48).

10. A recently published study (Houtz et al., 2021 in Molecular Ecology) testing many of the same hypotheses as this study should addressed in the intro and discussion.

This paper is now addressed in the Discussion (lines 268-271).

11. One smaller thing that might be helpful to add to the introduction - while vertical transmission in mammals might be rare, it seems that microbiome heritability may be high (see recent work here).

The microbiome heritability reported in the paper cited by the reviewer is 0.07, which is extremely low. Whether there is, or isn't, vertical transmission, it would not be possible to predict an association between microbes and hosts based this level of heritability. On account of its small contribution to the overall patterns of relationships, we do not believe it helpful to distinguishing among the varied hypotheses.